

# The history of mesowear: a review

Nicole L. Ackermans

Clinic for Zoo Animals, Exotic Pets and Wildlife, Vetsuisse Faculty, University of Zurich, Zurich, Switzerland

## ABSTRACT

Published mesowear data was reviewed from the year 2000 to November 2019 (211 publications, 707 species, 1,396 data points). Mesowear is a widely applied tooth wear technique that can be used to infer a herbivore's diet by scoring the height and sharpness of molar tooth cusps with the naked eye. Established as a fast and efficient tool for paleodiet reconstruction, the technique has seen multiple adaptations, simplifications, and extensions since its establishment, which have become complex to follow. The present study reviews all successive changes and adaptations to the mesowear technique in detail, providing a template for the application of each technique to the research question at hand. In addition, the array of species to which mesowear has been applied, along with the equivalent recorded diets have been compiled here in a large dataset. This review provides an insight into the metrics related to mesowear publication since its establishment. The large dataset overviews whether the species to which the various techniques of mesowear are applied are extant or extinct, their phylogenetic classification, their assigned diets and diet stability between studies, as a resource for future research on the topic.

## INTRODUCTION

Tooth wear can be measured on different physiological scales, from the microscopic (2D microwear (*Walker, Hoeck & Perez, 1978*) and 3D dental microwear texture analysis (*Schulz, Calandra & Kaiser, 2013a*; *Calandra & Merceron, 2016*; *DeSantis, 2016*; *Green & Croft, 2018*)) to the macroscopic (mesowear, absolute wear (*Fortelius & Solounias, 2000*; *Ackermans et al., 2019*)), informing us about a specimens', and by definition, a species' diet. Within tooth wear, attrition to the tooth's enamel surface caused by tooth-on-tooth contact is generally the main cause of wear in animals with a browsing diet. The soft nature of a browse-based diet causes opposing teeth to wear themselves, as the diet itself does not provide resistance (*Sanson, 2006*). Abrasion on the other hand, is caused by internal or external abrasives, which wear tooth material upon contact (*Janis, 2008*). Grasses contain large amounts of internal opaline silicates that wear tooth enamel when chewed repetitively (*Baker, Jones & Wardrop, 1959*), and grazing animals generally tend to feed close to the ground in open habitats, where plants become covered in external abrasives, for example dust and grit (*Janis & Fortelius, 1988*). It is still debated whether tooth wear is mainly caused by phytoliths (*Xia et al., 2015*; *Merceron et al., 2016*) or external abrasives (*Healy, 1967*; *Sanson, Kerr & Gross, 2007*; *Damuth & Janis, 2011*; *Hummel et al., 2011*; *Lucas et al., 2013*), and which is the main driver in the evolution

Corresponding author
Nicole L. Ackermans,
nlackermans@gmail.com

of hypsodonty, though the general agreement is that both types of abrasives contribute at least somewhat to tooth wear and thus to the evolution of hypsodonty (*Williams & Kay, 2001*; *Kaiser et al., 2013*).

Historically, tooth wear patterns have also been of interest for age determination, based on the visual aspect of the tooth's surface (*Grant, 1982*), using a technique that has been called 'macrowear' (*Pinto-Llona, 2013*), and confusingly, 'mesowear' (*Gonzalez-Socoloske et al., 2018*). It is important to note that, while this 'macrowear' is a species-specific technique—applicable to a variety of species from bears (*Stiner, 1998*) to manatees (*Gonzalez-Socoloske et al., 2018*)—this technique is solely applied to estimate age based on wear, and does not provide information on diet.

In the current review, mesowear is referred to as a series of techniques using a semi-quantitative method to evaluate tooth wear visible on the tooth profile with the naked eye, or handheld magnification. The original mesowear technique, or 'mesowear I', was introduced by *Fortelius & Solounias (2000)* (Table 1), as a method to reconstruct general palaeodiets of fossil ungulates by observing the macroscopic wear on their molars, specifically the M2. As such, in terms of dietary signal duration, mesowear serves as a midway point between the unworn shape of a tooth representing a general diet at an evolutionary scale (i.e. herbivore or carnivore), and microscopic wear, representing a specimen's last few meals (*Grine, 1986*). An abrasion-attrition wear gradient is used to assign dietary categories to herbivores, with browsers generally showing a more attrition-based wear pattern, and grazers a more abrasion-dominated pattern (*Fortelius & Solounias, 2000*). At the establishment of the technique, selenodont- (i.e. cow) or trilophodont-type (i.e. mastodon) molars were the target teeth for mesowear, applied by observing '*the buccal edges of the paracones and metacones of upper molars*' (*Fortelius & Solounias, 2000*) with the naked eye or at low magnification (*Fortelius & Solounias, 2000*, Fig. 1). As a direct consequence, mesowear is a fast, inexpensive technique for diet determination. Molar cusp relief or occlusal relief (OR) is defined in the original publication as '*the relative distance between cusp height and inter-cusp valleys*' (*Fortelius & Solounias, 2000*), with low OR related to the high abrasion typical of the grazer diet. Cusp shape (CS) is therein defined by '*the apex of the cusp being described as sharp, rounded or blunt*' (*Fortelius & Solounias, 2000*), using the maxillary M2 as the tooth of reference. Applying these variables allows dietary reconstruction based on the percentage of sharp, round, or blunt cusps; alongside the percentage of high relief. Mesowear I was developed using a database of 64 extant species (Supplemental Data), and was succinctly applied to six fossil species of known diet to test its strength, followed by a blind test on 20 specimens of *Hippotherium* (*Kaiser et al., 2000*) (Table 1; Supplemental Data).

In the original mesowear method described above, the sharper of the two molar cusps was scored on a wide variety of taxa, noting that the choice of cusp was not critical. This hypothesis has been confirmed by *Ackermans et al. (2018)* in a feeding experiment on goats, though significant inter-cusp differences have been detected in rhinoceroses (*Taylor et al., 2013*) and certain equids (*Taylor et al., 2016*). *Fortelius & Solounias (2000)* also note the importance of scoring at least 10- and ideally 20–30 specimens per species and/or locality for a reasonable approximation of the score, though on palaeontological

**Table 1 Additions and adaptations to the original mesowear technique-ordered by mesowear technique and date.**

| Technique | References | Description | Scores |
|---|---|---|---|
| Original mesowear—Mesowear I | *Fortelius & Solounias (2000)* | –Using the naked eye or ×10 magnification | OR: low, high |
| | | –Scoring only sharpest buccal cusp of maxillary M2 | CS: blunt, round, sharp |
| | | –Last molar in occlusion and M1 shape similar to M2 | |
| | | –Percentage of high relief and Percentage of sharp, round and blunt cusps | |
| Mesowear I—Adapted for Equidae | *Kaiser & Fortelius (2003)* | Method extended to all apices on maxillary P4–M3 in equids | Replaces original mesowear |
| Mesowear I | *Franz-Odendaal & Kaiser (2003)* | Method extended to maxillary M3, and mandibular M2 in ruminants | Replaces original mesowear |
| Mesowear I—Adapted for Lagomorpha | *Fraser & Theodor (2010)* | 'Cusp relief' combined with 'buccal shearing crush wear' on maxillary and mandibular P4–M2–resulting in 5 dietary classes | 1: 45° enamel-dentine relief with no additional wea<br>2: 45° enamel-dentine relief with buccal shearing crush wear<br>3: 45° enamel-dentine relief with buccal shearing crush & phase II wear<br>4: 90° enamel-dentine relief with no additional wear<br>5: 90° enamel-dentine relief with buccal shearing crush wear |
| Mesowear I—Adapted for Conodonta | *Purnell & Jones (2012)* | Scored on P1 elements | Not truly mesowear, does not have scores |
| Mesowear I—Adapted for Leporines & Murines | *Ulbricht, Maul & Schulz (2015)* | Classical mesowear on the maxillary M1–M2, and mandibular p3 in *leporinae* and distal side of the maxillary M1 and mandibular m1 in *murinae* | Same scores as original mesowear |
| Mesowear I—Adapted for voles | *Kropacheva et al. (2017)* | Maxillary M1–M2, mandibular m1 | Occlusal relief 1–7<br><br>Lateral facet development 1–3 |
| Mesowear II—'Mesowear ruler' | *Mihlbachler et al. (2011)* | Simplified score using gauges and a seven-point system | Combined score 0–6 |
| Mesowear II—'Mesowear ruler' | *Wolf, Semprebon & Bernor (2012)* | Additional intermediate scores | Combined score 0–13 in increments of 0.5 |
| 'Mesowear angles'—Adapted for Proboscidea | *Saarinen et al. (2015)* | *'Mean mesowear angles of three central lamellae in occlusion'* on all except deciduous teeth | Mean mesowear angle<br><106°: C3-plant based diet<br>>130°: C4-plant based diet (grazer) |

(Continued)

| Table 1 (continued). | | | |
|---|---|---|---|
| **Technique** | **References** | **Description** | **Scores** |
| "Mesowear angles"– Adapted for Xenarthra | *Saarinen & Karme (2017)* | All molariform teeth | For *Xenarthra, Folivora*: Mean mesowear angle: 60°–85°: fruit browsers 75°–100°: leaf browsers 100–132°: mixed-feeders 132°–150°: grass dominated mixed-feeders 150°–190°: grazers For *Xenarthra, Cingulata*: 60°–100°: carnivore, insectivore, omnivore, possibly browsers 100°–125°: browse-dominated mixed-feeders & herbivorous omnivores 125°–152°: grass-dominated mixed-feeder 152°–190°: grazers |
| Mesowear II | *Mihlbachler & Solounias (2006)* | Simplified score, only proportion of sharp cusps | Proportion of sharp cusps: 40–100%: Clean browser 20–40%: Mixed feeders: 0–20%: Grazer |
| Mesowear II 'quantitative mesowear' | *Widga (2006)* | Interval measurements of cusp and saddle heights to calculate cusp relief | Index of cusp relief: Low ICR: grazer High ICR: browser |
| Mesowear II | *Rivals & Semprebon (2006)* | Simplified score combining OR and CS | 0: high relief & sharp cusps 1: high relief & round cusps 2: low relief & round cusps 3: low relief & blunt cuspss |
| Mesowear II | *Kaiser (2009)* | | 0: high relief & sharp cusps 1: high relief & round cusps 2: low relief & sharp cusps 3: low relief & round cusps 4: low relief & blunt cusps |
| Mesowear II | *Rivals, Schulz & Kaiser (2009)* | | 0: high relief & sharp cusps 1: high relief & round cusps 2: low relief & round cusps 2.5: low relief & sharp cusps 3: low relief & blunt cusps |
| Mesowear II | *Croft & Weinstein (2008)* | | 0: high relief & sharp cusps 1: high relief & round cusps 2: low relief & round cusps 2.5: low relief & sharp cusps 3: high/low relief & blunt cusps |
| Mesowear II | *Fraser et al. (2014)* | Method extended to mandibular P4–M3 for ruminants | 1: high relief & sharp cusps 2: high relief & round cusps 3: high relief & very round cusps 4: low relief & round-blunt cusps 5: low relief & flat-blunt cusps |

| Table 1 (continued). | | | |
|---|---|---|---|
| **Technique** | **References** | **Description** | **Scores** |
| Mesowear II—Adapted for Marsupialia | *Butler, Louys & Travouillon (2014)* | Use of classical mesowear and a combined score on the maxillary left maxillary molars, scoring sharpest buccal cusp | Combined score as in *Kaiser (2009)* |
| Mesowear I & II—Expanded | *Winkler & Kaiser (2011)* | Intermediate stages added to original and combined score | OR: low, high-low, high, high-high<br>CS: blunt, round-round, round, round-sharp, sharp<br><br>Combined score 1–17 |
| Mesowear I and II—Expanded, Adapted for Rhinocerotidae | *Taylor et al. (2013)* | Expanded version and combined score on maxillary P2–M2. | Combined score 1–11 |
| Mesowear III—'Inner mesowear' | *Solounias et al. (2014)* | Scores the second enamel band, using a stereo-microscope<br>Mesial side, distal side and junction point are scored separately | Enamel band wear states:<br>1: ideal browser<br>2–3: intermediate<br>4: ideal grazer<br>Junction point score 1–4 |

specimens, tentative dietary assumptions can be made using a single tooth (*MacFadden, 2009*; *Rivals et al., 2017*), as well-preserved specimens are rare. Although the initial assumption was that mesowear remains relatively stable throughout an individual's life (when very young or very old specimens are excluded), *Rivals, Mihlbachler & Solounias (2007)* later established the idea that mesowear varies based on initial crown height and can be different throughout an animal's lifetime. The age structure of samples measured by mesowear should thus be taken into consideration, especially in the case of brachydont species.

Further adaptations were made to the original mesowear technique (for more details, see Table 1), expanding it to more teeth (*Franz-Odendaal & Kaiser, 2003*; *Kaiser & Fortelius, 2003*), and adapting the method to specific taxa (*Fraser & Theodor, 2010*; *Purnell & Jones, 2012*; *Taylor et al., 2013*; *Butler, Louys & Travouillon, 2014*; *Saarinen et al., 2015*; *Ulbricht, Maul & Schulz, 2015*; *Kropacheva et al., 2017*; *Saarinen & Karme, 2017*). Some, deeming OR a redundant measure not fully independent from CS, simplified mesowear by only using categories of CS (*Mihlbachler & Solounias, 2006*; *Widga, 2006*), while others simplified the technique by combining OR and CS into a single score (*Rivals & Semprebon, 2006*; *Croft & Weinstein, 2008*; *Kaiser, 2009*). However, combining these scores can lead to oversimplification and, depending on the aim of the study, the possibility to isolate tooth height or sharpness could be crucial. These simplified versions of the original mesowear technique were deemed 'mesowear II' by *Solounias et al. (2014)*. Further simplifications include a 'mesowear ruler' system (*Mihlbachler et al., 2011*), and a 'mesowear angle' system (*Saarinen et al., 2015*). Mesowear I and II also have an extended version, where intermediate stages were added to the original mesowear categories and

a more complex combined score was created to provide more detail (*Winkler & Kaiser, 2011*) (Table 1). 'Mesowear III' or 'inner-mesowear' was implemented by *Solounias et al. (2014)*, where scoring the inner enamel band of the tooth aimed to record a more precise signal and represent a shorter timeframe. Mesowear III has been applied in six other studies since it was established (Supplemental Data), but has been tested experimentally once, and results did not show more precision than traditional mesowear when both techniques were applied to the same dataset (*Stauffer et al., 2019*).

Traditionally, mesowear has either been scored directly on the specimens' teeth, on resin casts, or on photographs of the specimen's teeth (*Fortelius & Solounias, 2000*). More recent studies, however, have used 3D models of wear facets (*Hernesniemi, Blomstedt & Fortelius, 2011*), or scored mesowear directly onto 3D reconstructions from CT scanned skulls of live animals (*Ackermans et al., 2018*). Various microscopy techniques have also been used as a means of scoring mesowear on smaller specimens such as conodonts (*Purnell & Jones, 2012*), lagomorphs, and rodents (*Ulbricht, Maul & Schulz, 2015*; *Kropacheva et al., 2017*).

The many iterations and addendums to the original mesowear technique can create confusion regarding the category of mesowear best applied (*Viranta & Mannermaa, 2014*), and the interpretation of corresponding results (*Díaz-Sibaja et al., 2018*). The aim of this review was to therefore create a body of reference with precise definitions and short explanations for each variation of the mesowear technique, to facilitate future application. An overview of current dental wear techniques exists (*Green & Croft, 2018*), but the current study provides a more detailed and widely understandable overview of the history and progression of the mesowear technique in particular. For this purpose, Table 1 lists all major amendments to the original mesowear technique—including the various versions of mesowear I, II and III—along with a short description and the scoring system used, thus hoping to ease comprehension of the available techniques and promote comparability of studies. In addition, a dataset was created reuniting the dietary classifications of all species to which the mesowear technique has been applied thus far, including specimen type, phylogenetic classification, and diet, as a readily accessible resource for future research (Supplemental Data).

## METHODS

Publications were recorded using the search term 'mesowear' in Google Scholar ($n = 1{,}150$), PubMed ($n = 25$), ResearchGate ($n = 230$), and Web of Science ($n = 142$), for every year from 2000 until the present (11 November 2019). After removing duplicates and non-relevant studies (using the terms 'mesowear' or 'macrowear' to describe wear on the macroscopic scale, without referring to the *Fortelius & Solounias (2000)* mesowear technique), $n = 211$ publications were analysed. Book chapters, PhD, M.Sc theses, and conference proceedings were included if they contained otherwise unpublished original mesowear data.

Diets in the supplemental raw data are indicated as shown in the corresponding references (references for the Supplemental Data are in Annex 1). A 'various' diet indicates a diet change for the same species within the publication (different localities or time

periods). A species without an assigned diet represents the lack of a diet indication or mesowear score within the text. An 'experimental' diet represents studies in which experimental diets were fed to animals in controlled environments. When a study measured both mesowear and microwear (or another dietary proxy) and the indicated diets diverged, the diet determined by mesowear scoring was reported here. If species were listed with multiple entries for different localities, collections, or subspecies within a single study, an average was made. If within a study mesowear was scored but the diet was not defined, a diet was assigned according to the mesowear score reported in the publication and previous research regarding the respective technique. Extant and extinct specimens were classified as either 'wild', 'captive' (zoo or experimental specimens), archaeological (excavated in an archaeological context as defined by the original publication, designated 'extant_a' in the Supplemental Data) or fossil (fossil specimens of extant species designated 'extant_f' in the Supplemental Data). When a palaeontological specimen's identification could not be established to the species level, the specimen was designated as 'fossil' in the Supplemental Data. For simplicity of analysis, mesowear techniques are designated mesowear I, II, III, or a combination thereof in the Supplemental Data. Extended or simplified versions are only noted in the case of the 'mesowear ruler', 'mesowear angle', 'mesowear I and II—extended', and all taxon-specific techniques. Data was arranged using pivot tables in Microsoft Excel (version 16.26) for graphic representation and interpretation.

Although mesowear can vary within species at different localities or different points in time, the constancy of diets assigned to a species using mesowear was assessed using the dataset assembled in the present study. It should be noted that the extreme variability between publications makes this a very coarse measure, however, it may either serve as an indication of the consistency of a species' diet, or as an indication of the difficulty to consistently assign a score to the species. When species were scored in more than one publication, a simple metric was devised: within a species, the diet recorded by the highest number of publications was calculated as a percentage of the total number of publications measuring mesowear in this species (i.e. For moose, *Alces alces*, mesowear was scored in seven publications, of which six reported a browser diet. Thus, using this metric, the moose's main-diet percentage is 86% browser). This percentage was then plotted against the number of publications scoring the species—in this case a higher main-diet percentage, alongside a high publication count indicated a more robust diet. This was measured using the dataset from the Supplemental Data including all types of diets, as well as using a simplified version of this dataset excluding all but the 'grazer', 'browser', and 'mixed-feeder' diets (Fig. 1).

## RESULTS

The data collected (Supplemental Data) shows that, when ordering the data by publication, 55% of all publications score exclusively extinct specimens, while 17% apply mesowear to solely extant species. Five percent of publications score solely extant archaeological or fossil specimens (extant_a or extant_f); while 10% score a mix of extinct and extant specimens, the rest scoring combinations of the above (Fig. 2A). Only four publications

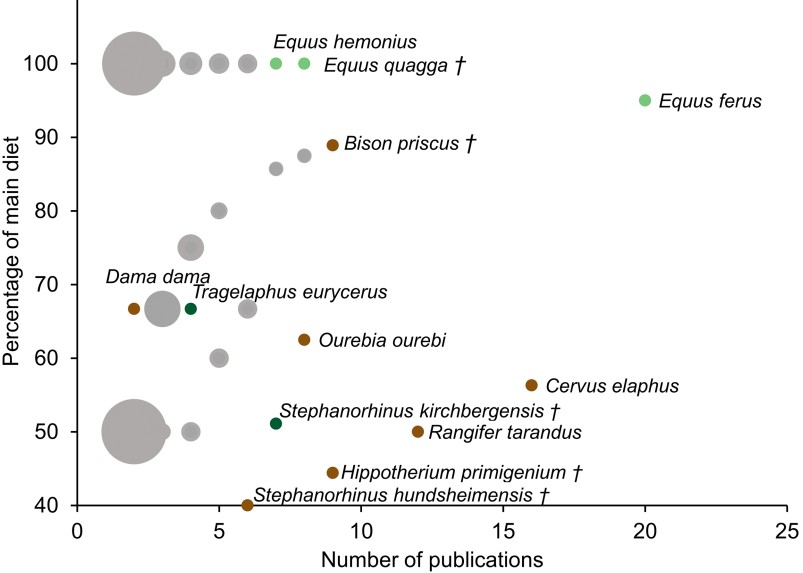

**Figure 1 Dietary robustness of species represented in a mesowear dataset from 2000 to November 2019.** Dietary robustness is a measure represented by the percentage of a species' main diet throughout publications, plotted against the number of publications featuring the species. Size of marker indicates the number of species per point (minimum 1, maximum 44). Markers are grey when multiple species occupy the same graph space (i.e. 41 species have the same diet when scored in two publications). Light green markers represent grazers, dark green markers represent browsers, and brown markers represent mixed diets, † indicates extinct species, *n* = 188 Publications.

applied mesowear to captive or experimental animals representing roughly 2% of all studies. With regards to diet, the mixed diet is most highly represented among all species (33%) as it covers a large spectrum, followed by the browser diet at 26% (Fig. 2B).

When ordering the data by technique and publication, 'mesowear I' on its own was scored in 37% of studies, followed by 'mesowear II' (21%), 'mesowear ruler' (14%), and 'mesowear I and II' (9%), the rest using a combination thereof, or taxon-specific techniques (Fig. 2C). Most taxon-specific techniques were only used once in their original publication, with the exception of 'mesowear adapted for Proboscidea', used in nine publications and 'mesowear adapted for Conodonta' used in four. This fits within a statement from the original mesowear study, stating that '*care should be taken not to lose the generality of the method, since restricting it to a single, morphologically uniform group will serve to limit the choice of recent species available for comparison*' (*Fortelius & Solounias, 2000*).

Out of the 211 publications analysed, 17 studies scored over 20 species, with the highest number of species being 85 (*Solounias et al., 2013*). Placental mammals were overwhelmingly scored (95%), though they were surprisingly not the only class of animals to which mesowear was applied. *Butler, Louys & Travouillon (2014)* adapted mesowear to marsupials, and *Purnell & Jones (2012)* applied mesowear to fossil conodonts (Table 1), a technique which was also applied to elasmobranches (*McLennan, 2018*). When sorted by order, artiodactyls were most represented (63%), followed by perissodactyls (26%) (Fig. 2D). Overall, out of 707 species (excluding 'sp.'), *Equus* was by far the most

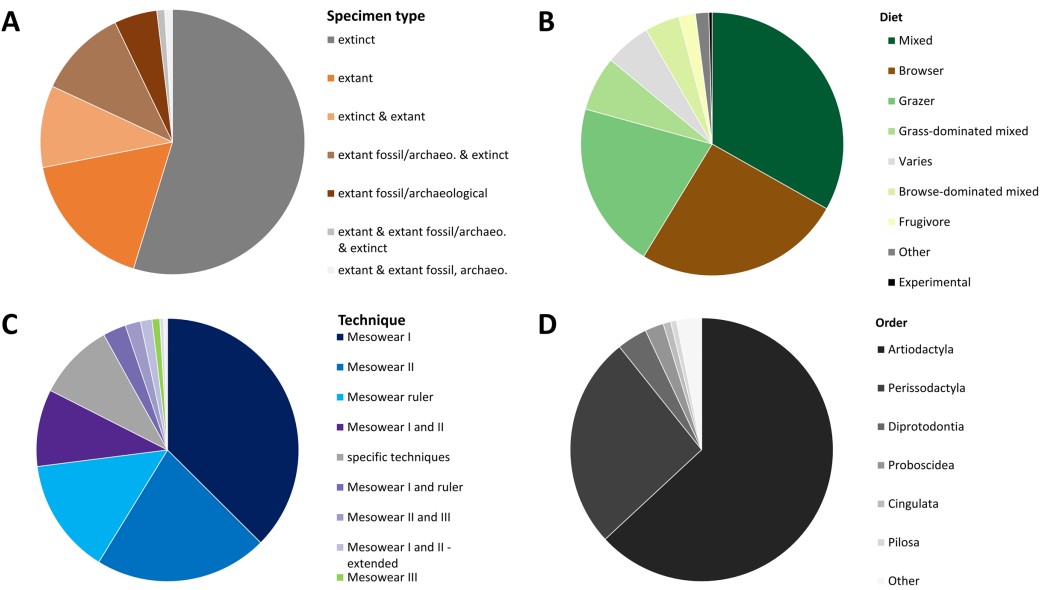

**Figure 2 Overview of a mesowear dataset from 2000 to November 2019.** (A) Specimen status of samples. (B) Percentages of diets. (C) Percentage of different mesowear techniques employed. (D) Percentage of taxonomic orders. *n* = 211 Publications.

scored genus, with 109 counts, followed by *Tragelaphus* with 38 counts, and *Cervus* at 37 counts. At the species level, *Cervus elaphus* was most commonly scored, with 19 counts, followed by *Equus ferus* (18 counts). In total, 177 species were scored in more than one publication, meaning that about 75% of species were only scored once (Supplemental Data).

In part because of the number of times it is represented in the dataset and because of its extreme hypsodonty, the species with the most robust unchanging diet is *E. ferus*, with 95% diet robustness within 20 publications (Fig. 1).

## DISCUSSION

Although one may envision more sophisticated or precise methods of palaeodietary reconstruction, it is important to remember that the original goal of the mesowear technique was to provide a fast and cost-effective way of determining diets for a large number of species. It has been thoroughly tested for this purpose and is extremely efficient in determining herbivore diets in a broad sense. The array of mesowear measurement techniques stemming from the original method have their respective pros and cons. If the technique is too simplified, we run the risk of hiding more subtle variations in diet. The 'mesowear ruler technique' was originally designed for use on horses (*Mihlbachler et al., 2011*) but was later applied to other species without adaptation or further tests of robustness (*López-García et al., 2012*; *Rivals, 2012*). Additionally, adapting the technique to species with very specific tooth morphology, such as proboscideans (*Saarinen et al., 2015*), adds the advantage of being able to score diets for these species, but this can only be reliably reached through copious amounts of testing. Fine-tuning mesowear to every taxon runs the risk of tarnishing the main goal of mesowear, that is being fast and cost efficient, and most importantly, the creation of so many techniques reduces comparability

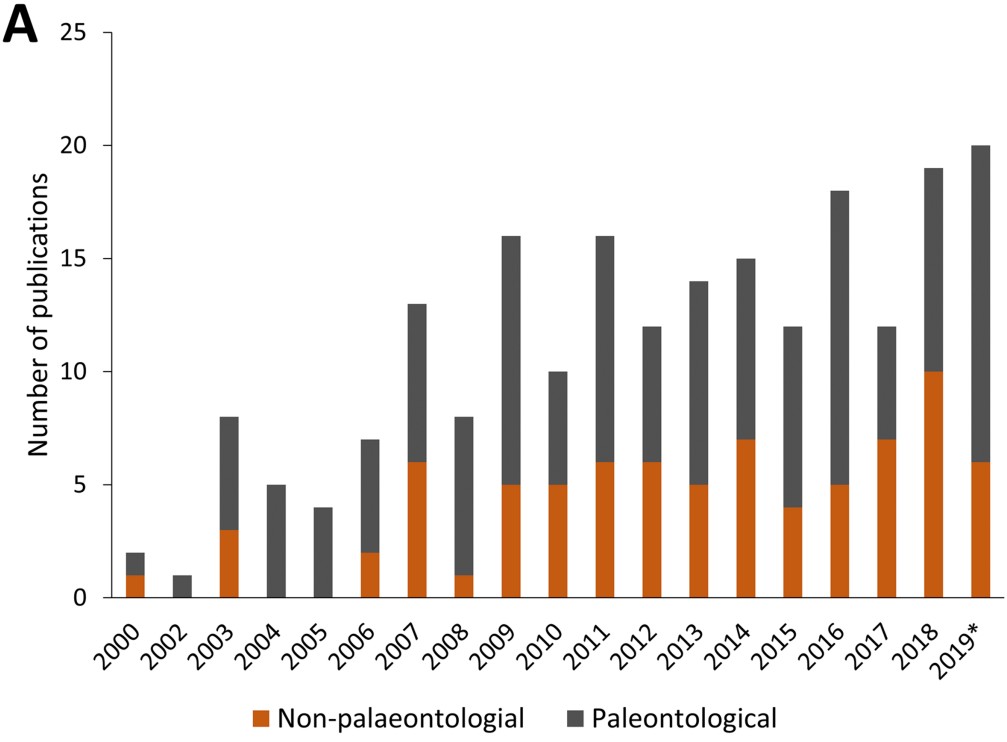

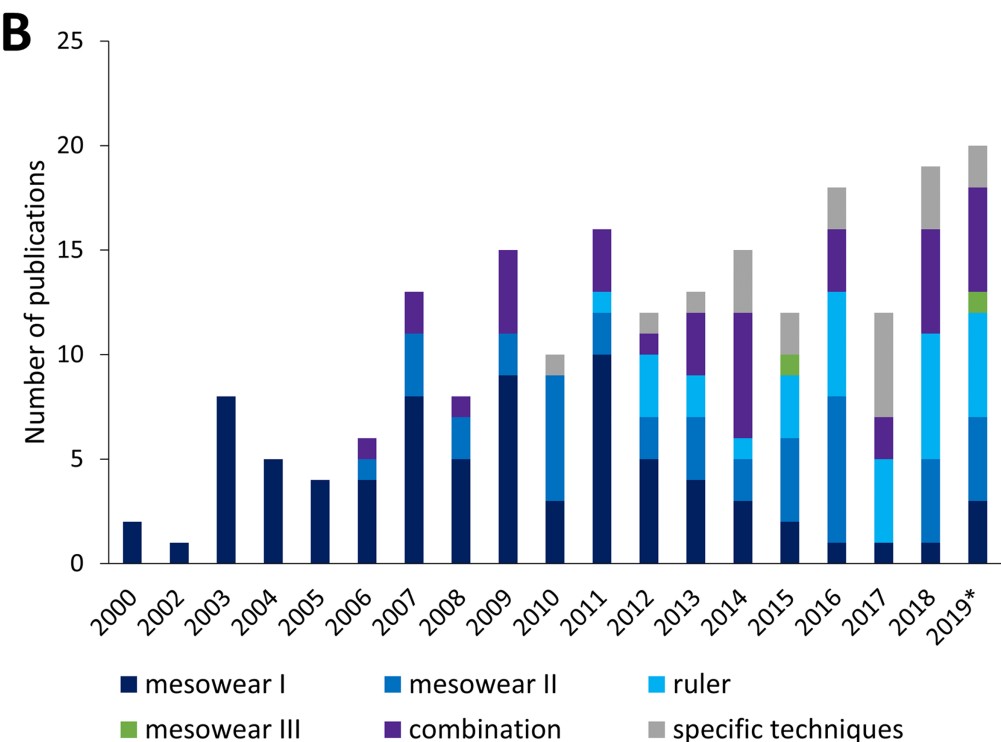

**Figure 3 Yearly amount of publications scoring mesowear between 2000 and November 2019.**
(A) Paleontological specimens vs. non-paleontological specimens. (B) Different techniques used to score mesowear, data sorted by number of publications. $n$ = 211 Publications. *Not a complete year.

between studies. Ideally, if the majority of studies applied the extended version of mesowear (*Winkler & Kaiser, 2011*), from which mesowear I scores can be easily deduced, this would enable higher comparability between studies, all while remaining a quick and easy technique. Based on the present review, it is suggested to apply only the extended mesowear technique in future studies. It can be measured as fast as any other technique using a simple guide (*Taylor et al., 2016*, Fig. 1), and extended mesowear I scores are easily converted into either the conservative mesowear I scores, or extended and conservative mesowear II scores (*Winkler & Kaiser, 2011*, Table 3). Thus, this technique allows flexibility as well as reproducibility between studies, providing a more cohesive body of work in the future.

The 'dietary robustness measure' established here may be coarse, but it provides a different approach in investigating dietary robustness. It also represents the number of species scored within a single publication, demonstrating some species have been scored in over 10 separate publications. Because there is no independent proxy for diet, this metric cannot discriminate between stability of diet and reproducibility of the method itself, though these are aspects that are relevant to future development of the methodology, as some mesowear studies could simply be interpreting the same mesowear data from a different perspective. However, providing an overview of the variability in mesowear scoring may allow for a re-balance of mesowear application in future studies, by increasing reproducibility and reducing repeated measures, for example on species with high dietary robustness.

Since the creation of the mesowear technique, the number of publications per year, as well as the type of publication (palaeontological or not) has grown until around 2010, with a roughly even distribution between non- and purely-palaeontological publications (Fig. 3A). The type of mesowear technique applied over the years also varies, and the number of publications applying solely 'mesowear I' appears to decline over time as it becomes part of a combination of techniques, while the use of taxon-specific techniques increases (Fig. 3B). Mesowear remains an essential asset for dietary reconstruction and has become more frequently applied in combination with other dietary proxies such as microwear or isotopic data, to provide a more accurate representation of diet over different timescales, though these proxies are rarely in accordance, and the development of wear on different scales remains to be investigated (*Ackermans et al., 2020*).

A precise understanding of dietary timescales requires the establishment of a baseline, to be used as a reference in defining the length of a dietary signal. In the case of mesowear, very few publications investigate mesowear experimentally (*Solounias et al., 2014*; *Kropacheva et al., 2017*; *Ackermans et al., 2018*; *Stauffer et al., 2019*) due to the cost and time required for long-term animal experiments. Because of this, the duration of the dietary signal represented by mesowear remains widely unknown. The few experimental tests of mesowear that can be considered long-term seem to confirm this proxy as representing more of a general lifetime signal, at least in small ruminants (*Ackermans et al., 2018* on goats for 6 months; N.L. Ackermans, L.F. Martin, D. Codron, J. Hummel, P.R. Kircher, H. Richter, M. Clauss, J-M. Hatt, 2020, in preparation on sheep for 17 months). However, it is impossible to experimentally recreate the variations of nature, and the comparison of the aforementioned results to those where mesowear shows more seasonal effects (*Kaiser & Schulz, 2006*; *Schulz et al., 2013b*; *Marom, Garfinkel & Bar-Oz, 2018*)

requires further investigation. Further areas of development for the mesowear technique could also explore objectivity through automation using image processing algorithms or artificial intelligence (*Karme, 2008*), though these methods are time-consuming to develop. A better understanding of the timescale represented by mesowear can only improve the precision of dietary reconstructions, all while furthering our understanding of the dental wear and dietary habits of extant species.

## ACKNOWLEDGEMENTS

I thank Marcus Clauss for input, ideas, and insightful corrections, as well the two reviewers, Florent Rivals and Ivan Calandra for their helpful and constructive comments.

### Funding

This study was part of project 31003A_163300/1 funded by the Swiss National Science Foundation (Schweizerischer Nationalfonds zur Förderung der Wissenschaftlichen Forschung). The funders had no role in study design, data collection and analysis, decision to publish, or preparation of the manuscript.

### Grant Disclosures

The following grant information was disclosed by the authors:
Swiss National Science Foundation (Schweizerischer Nationalfonds zur Förderung der Wissenschaftlichen Forschung): 31003A_163300/1.

### Competing Interests

The author declares that they have no competing interests.

### Author Contributions

- Nicole L. Ackermans conceived and designed the experiments, performed the experiments, analysed the data, prepared figures and/or tables, authored or reviewed drafts of the paper, and approved the final draft.

### Data Availability

   The dataset is available in the Supplemental Files.

### Supplemental Information

Supplemental information for this article can be found online at http://dx.doi.org/10.7717/peerj.8519#supplemental-information.

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
