# Peer review of "The history of mesowear: a review"

_PeerJ, doi:10.7717/peerj.8519_

## Round 0.1 · original submission · Minor Revisions

I think the reviewers made good points that I agree with. If you follow or comment on their points, thus make the minor revisions, the manuscript will be considered for publication.

·

Basic reporting

The manuscript is clear and unambiguous. It provides an extensive review of the literature on mesowear over the past 20 years. It summarize the history of mesowear and its evolution with the description of the different methods that were designed.
The literature references are exhaustive and I could not identify any missing work.

It is very useful to have access to the raw data. Besides providing the original data, it is also a source of information to find mesowear data for species or taxa.

Minor corrections:

Lines 46-46. Who did use “meso-wear” to refer to age determination technique? I checked Grant (1982) and I did not see it. Grant is using “MWS” but not meso-wear. Please add the references where “meso-wear” is employed.

Line 84. The reference Rivals et al. (2007a) is missing from the list. Additionally, there is no “2007b” in the text, so it should be cited as (2007).

Line 85. At the end of the sentence, I would suggest to add “lifetime, especially for brachydont taxa” because the signal is much more stable in hypsodonts.

Line 92: The two variables, OR and CS, were combined not only to simplify, but because the two variable are not fully independent. They convergence towards the grazing end of the dietary spectrum, i.e. blunt teeth have low relief. This is important when the two variables are used for statistical tests.

On Figs. 4, 5, 7, 9 it would be useful to have the N on the figure or the caption. To know how many papers belong to each category.

Captions for Fig. 2 and 3 are inverted in the pdf.

The list of references in Annex 2 needs to be checked carefully as I could spot some mistakes:
- Many Latin names of species are not italicized.
- What are the papers cited at the beginning of the list?
- The reference Diana P, Juha S……. (2020) is wrong. These are the first names of the authors. The correct list of authors is: Pushkina, D., Saarinen, J., Ziegler, R., Bocherens, H., 2020.

Experimental design

The methods are well explained and I could not identify any missing information. Most of the sources are adequately cited (see minor mistakes in section 1).

Validity of the findings

The finding are valid and provide important points for future research on mesowear.

·

Basic reporting

This manuscript is a welcome review on mesowear analysis. As rightly stressed by the author, there are now currently many variations of the original method and it is difficult to get a good overview of the pros, cons and field(s) of application of each variation. This manuscript does just that, in a clear and concise manner.
The manuscript is well-written: the structure is logical and the wording is clear.

The intro and the background do explain the relevance of the study, and reference to previous work adequately.
However, two topics have been excluded from this review, and I believe they deserve some space:
- Mesowear as an environmental proxy - there have been quite a lot of discussion and debate around the signal provided by mesowear: is it a dietary or an environmental proxy? I do not think this debate is closed, but in any case, this topic must be addressed in the introduction.
- The author does not provide enough background and clarity regarding the different scales of wear analyses and how they relate to each other: how microwear affects mesowear and how both affect hypsodonty? Are the different methods just about different dietary signal durations, or are the wear agents and processes different at different scales? There is a lot at play here, and the few sentences on this topic (lines 40-45 and 230-232) are too vague and are mixing many concepts in an ambiguous manner. Please expand on this topic, in the introduction and discussion, potentially linking it to my previous comment.

Raw data are shared as appendices.

The figures are OK, but I am one of those persons having a strong opinion against pie charts, especially with so many categories (e.g. Fig. 4 is simply unreadable). A simple Google search “pie charts why not” will provide a lot of clear arguments why pie charts are most of the time suboptimal and should be replaced with bar charts. Some might say it is a matter of taste, but this is only partly true. Hopefully the author and/or editor will be convinced by all the blog entries and articles discussing the topic, and the revised version will replace pie charts with bar charts.

Experimental design

The methodology is described in enough details (pending some minor comments – see “General comments”).
The review is, as far as I can tell, exhaustive and unbiased. The sources are correctly cited. There are just a few places where pages are not cited for quotations (see “General comments” – comments #16 & 31).

Validity of the findings

I think the results are interesting per se. But I also believe that the author could make more of it, provide recommendations for current use, as well as more avenues for future improvements (see “General comments” – especially comments #19-20, 33-34, 37).

Additional comments

In this section, I detail some minor comments, following the manuscript’s structure.

ABSTRACT
1) Line 10: I am not a supporter of the royal “we”. This is more stylistic than anything, but I would recommend simply writing “I” in this sentence. Passive voice is also OK.
2) Line 16: repetition of “present”; maybe “The present study reviews in detail…”

INTRODUCTION
3) Line 28: typo in “Introduction”.
4) Line 30: although not everybody agrees, the most common term for the 3D microwear method is “dental microwear texture analysis” or “DMTA”. Rather than two out of many case studies, I think that citing reviews will be more suitable here (e.g. Ungar 2015, Calandra & Merceron 2016; DeSantis 2016; Green & Croft 2018 for some recent ones).
5) Line 32: “… about a specimen’s and, by extension, a species’ diet”.
6) Lines 40-45: some references are switched between the “phytolith” (Xia et al. 2015, Merceron et al. 2016…) and “grit” (Lucas et al. 2013, Sanson et al. 2007…) proponents. Please check and correct.
7) Lines 45-50: this topic deserves its own paragraph.
8) Line 45: “… tooth wear patterns have also been…”
9) Lines 49-50: “… this technique is solely applied to estimate age…”
10) Lines 51-52: mesowear is not only performed with the naked eye, as shown in Table 1 and on lines 105-110. Please clarify here.
11) Lines 53-55: I recommend merging this sentence with the sentence on lines 57-59; it is a bit repetitive as is.
12) Line 55: the original method is applied only to the M2 (Table 1).
13) Lines 55-57: I recommend moving this sentence to lines 66-71. This will at the same time help explaining to the non-expert what “attrition-based/abrasion-dominated wear patterns” are.
14) Line 59: replace “length” with “duration”.
15) Line 60: “… at the evolutionary scale…”.
16) Lines 63-64: the quotation is missing a reference (including page number).
17) Line 79: it is not clear who “the authors” are here.
18) Line 82: please add references to support the claim that “rudimentary dietary assumptions are sometimes made using a single tooth”.
I think the wording “as complete specimens are rare” is somewhat misleading. To me, a “complete” specimen would be a full skeleton/skull/mandible…; I guess “well-preserved specimens” is more appropriate here. But my interpretation might be wrong.
19) Lines 82-85: does it then mean that the diet is not stable through the lifetime? If it is indeed not, then it has huge implications for mesowear analysis. Please expand.
20) Line 90: OR seems redundant because it is correlated to CS. But are there exceptions, or in other words, how good is the correlation? I think this would be the space for a critical, rather than purely descriptive, review.
21) Lines 96-98: it is not clear to me whether this is mesowear I or II. Based on Table 1, it is apparently both, but I do not see how it can be both… Is it just that the authors extended and applied both methods at once?
22) Line 121: “promote enhance”; this is redundant, probably a typo.

METHODS
23) Line 127: maybe “recorded” is a more appropriate word than “cited” here.
24) Line 131: “… n=211 publications were analysed.”
25) Line 132: “… PhD and MSc theses…”
26) Lines 140-141: how can species be “listed with multiple entries within a single study”, if it is not because of localities and time periods (which would mean “various” diet)?
27) Lines 145-146: I find the terms “extant*-**” make it difficult to remember what they mean. What about “extant_archaeo” and “extant_fossil”, or something similar?
28) Line 159: I do not understand what the “percentage of the species’ main diet” is. Can you please elaborate here?

RESULTS
29) Lines 166-169: refer to the naming used in Fig. 5a, so that the link between text and figure is easier to follow. See also comment #27.
30) Lines 171-173: there is a simple explanation for this pattern: categories other than browsing/grazing/mixed are taxon or paper-specific.
31) Lines 180-182: add a reference, with page, for this quotation.

DISCUSSION
32) Line 202: please define what you mean with “coarse” scale.
33) Lines 212-215: is this a recommendation for future studies? Is this the methodology that seems the most promising, based on the present review? If yes, I recommend emphasizing it, and expanding on it (including the possible limitations of using this methodology over any other, or of using only one methodology…). As it is, this review is mostly purely descriptive; I argue this here would be the missing synthesis.
34) Lines 216-217: because there is no independent proxy for diet, this metric cannot discriminate between stability of diet and reproducibility of the method itself (as mentioned lines 155-157), both aspects being very relevant to the future development of the methodology. However, this ambiguity is somewhat forgotten in the discussion. The reproducibility aspect should be stressed again here, interpreting the same data from a different perspective.
35) Line 239: “… seem to confirm…”
36) Another avenue for future development concerns automation and objectivity of the method. This relates somewhat to comment #34. I know that some have tried to automatically score mesowear from photographs, using image processing algorithms. Yet, as far as I know, no automatic method has been proposed, probably because the gain is not worth the time needed to develop it. Still, this remains a potentially very important development for future studies, especially for “increasing reproducibility and reducing repeated measures” (lines 220-221).

TABLE and FIGURES
37) Table 1:
- The table could be made much more useful/practical in restricting the column “scores” to include only true scores (like for the original mesowear, unlike e.g. Mesowear I - Adapted for Lagomorpha), potentially giving more details (e.g. Mesowear II “quantitative mesowear”), and adding a column “diet” or “interpretation” of the scores.
- Additionally, I think some methods have been replaced by newer ones. Therefore, an extra column about which method replaces which one could make this table even more useful.
- It should be explained why the method Mesowear I – Adapted for Conodonta does not have scores (i.e. it is not truly mesowear).
38) Fig. 1 (legend):
- I recommend moving “from Taylor et al. (2016)” to the end of the legend.
- Capra hircus has no mention of cusp height (hypsodont, right?)
- Never heard the term “hyperhypselodont”. A horse tooth is not hypselodont (ever-growing) anyway, even though the root forms late during ontogeny. Maybe “hyperhypsodont”?
39) Figs 2-3: I think the legends have been switched.
40) Fig. 7b: I do not understand what the author means with “Grey markers indicate multiple species (and multiple diets)”, which also means I do not understand what the bulk of the figure shows. I also recommend choosing another pattern (color) for mixed diets.
41) The numbering of the figures does not follow the order in the main text. I recommend some reorganization:
- Figs 5a+7a+6a+4 --> new Fig. 4
- Figs 5b+6b --> new Fig. 5
- Fig. 7b --> new Fig. 6

I hope these comments are constructive and will help improving this manuscript.

---

## Round 0.2 · accepted · Accept

You reacted to all comments of the reviewrs and with the minor revision it is well acceptable for publication and represents an interesting review of mesowear.